# Cross-Medium Photoacoustic Communications: Challenges, and State of the Art

**DOI:** 10.3390/s22114224

**Published:** 2022-06-01

**Authors:** Muntasir Mahmud, Md Shafiqul Islam, Akram Ahmed, Mohamed Younis, Fow-Sen Choa

**Affiliations:** 1Department of Computer Science and Electrical Engineering, University of Maryland Baltimore County, Baltimore, MD 21250, USA; mdislam1@umbc.edu (M.S.I.); choa@umbc.edu (F.-S.C.); 2Department of Computer Engineering and Center for Communication Systems and Sensing, King Fahd University of Petroleum & Minerals, Al Dhahran 31261, Saudi Arabia; akram@kfupm.edu.sa

**Keywords:** photoacoustic, optoacoustic, air-to-underwater communication, cross-medium communication, underwater communication

## Abstract

The current era is notably characterized by the major advances in communication technologies. The increased connectivity has been transformative in terrestrial, space, and undersea applications. Nonetheless, the water medium imposes unique constraints on the signals that can be pursued for establishing wireless links. While numerous studies have been dedicated to tackling the challenges for underwater communication, little attention has been paid to effectively interfacing the underwater networks to remote entities. Particularly it has been conventionally assumed that a surface node will be deployed to act as a relay using acoustic links for underwater nodes and radio links for air-based communication. Yet, such an assumption could be, in fact, a hindrance in practice. The paper discusses alternative means by allowing communication across the air–water interface. Specifically, the optoacoustic effect, also referred to as photoacoustic effect, is being exploited as a means for achieving connectivity between underwater and airborne nodes. The paper provides background, discusses technical challenges, and summarizes progress. Open research problems are also highlighted.

## 1. Introduction

Recent years have witnessed growing interest in the use of networks in underwater applications, such as environmental state monitoring, search and rescue, marine biology, leak detection in oil fields, tracking seismic activities, seabed profiling, distributed surveillance, and navigation [1,2,3,4,5,6,7,8,9,10,11,12,13,14,15,16,17,18,19,20]. While lots of research has focused on establishing robust communication links to enable reliable exchange of information among the underwater nodes, little attention has been given to interfacing such an underwater network to terrestrial-based centers. Fundamentally, it is often required for the data collected by underwater nodes to be relayed to a remote off-water command post. However, interconnecting underwater nodes with terrestrial assets have to be provisioned. Basically, the terrestrial systems utilize Electromagnetic waves (EM) for communication while EM suffers from high attenuation loss in the underwater environment. Moreover, acoustic signals which are deemed the best means for underwater communication [21,22,23] cannot cross the air–water interface due to the high impedance mismatch between the two mediums.

To overcome such an interconnection issue, published underwater networking solutions incorporate floating nodes, e.g., buoys, to act as bridges between the acoustic and EM networks and as reference anchors for underwater localization. However, in many application scenarios deploying surface nodes could be unjustified due to the incurred cost relative to the task duration, or undesirable due to mission constraints. For example, in search and rescue missions, e.g., finding the black box of a crashed plane, autonomous underwater vehicles (AUVs) survey the area for a limited duration. In such a case, requiring the deployment of surface nodes is logistically challenging and very slow. The search for the Malaysia Airlines Flight 370 in 2014 demonstrated how tardiness could lead to failure; basically, the water current and the uncertainty about the flight path complicated the process where boats are involved in the search, mainly to deploy probes and interact with AUVs. Other examples include, but are not limited to, scientific data collection and covert military underwater surveillance where the presence of the underwater network has to be concealed.

In addition, the design of underwater acoustic networks (UANs) has its own challenges [24,25]. First, nodes tend to drift, causing some communication links to be lost and the network to get partitioned. Moreover, the inhomogeneity of the underwater environment causes acoustic waves to travel with varying speeds that depend on the temperature, density, and salinity of the water, which complicates both link and network formation [25,26,27,28]. Therefore, UANs ought to adapt to changing seawater conditions and the inherent node mobility in order to reestablish broken communication links. The use of floating nodes has been conventionally used to facilitate management of the UAN topologies, where the floating nodes can serve as anchor nodes with known GPS coordinates to guide the underwater nodes. In other words, the floating nodes not only interface the UAN to airborne and terrestrial command centers and base-stations, but also generate acoustic beacons to help underwater nodes determine their relative position using multi-literation [29,30,31,32,33].

To address the aforementioned networking challenges, there is a need for developing network architecture and protocols that take into consideration the underwater physical layer dynamics and the constraints on interfacing the network with off-water nodes. Given the transmissivity of light from air to water is very high, visible light, e.g., LED, and laser could be a viable candidate for cross medium communication [34,35]. However, neither LED nor laser light is good for long range communication due to the high light absorption and beam scattering, especially for a wavy water surface, which is the most common scenario. For long distance communication from air to underwater optoacoustic (photoacoustic) energy conversion is indeed a viable option. Optoacoustic energy conversion means converting light energy into acoustic energy. Such an energy conversion mechanism was discovered long ago by Alexander Graham Bell in 1881 [36]. Basically, an aerial vehicle would transmit a laser beam towards the surface; the beam would then penetrate the water surface and create an acoustic wave inside the water. Such a generated acoustic wave will propagate in the underwater medium and face the classical challenges of spreading, attenuation, and multipath fading. Hence, contemporary channel estimation techniques that take in consideration fading and other underwater characteristics are often applied [37,38]. This paper focuses on challenges faced at the air–water interface, particularly on issues related to optoacoustic communication.

An optoacoustic link can be established by controlling the properties of the laser beam to carefully modulate the corresponding acoustic wave with the desired data. An example application scenario of optoacoustic communication is illustrated in Figure 1, where an airborne vehicle is assisting an underwater network in localizing nodes and forming a global coordinate system. The airborne vehicle instantiates acoustic emitters at multiple points on or close to the water surface through the optoacoustic effect. These points act like underwater or surface nodes. By carefully modulating the laser beams, the generated acoustic waves could be demodulated to infer the GPS coordinates of the position of the emitters (points). The underwater nodes that are within the range of the generated acoustic signal will localize themselves and then attempt to localize other nodes at further depth. Such localization will be critical for performing certain missions such as finding a black box of a crashed plane, where accurate navigation and target position estimation are achieved. As the underwater nodes move as part of the search for the black box or drift due to water current, the aerial unit will repeat the process to sustain robustness.

Photoacoustic-based communication is deemed as a game-changer in the realm of sea-based networks; no wonder it is an emerging area of research. A number of fundamental issues are being addressed related to acoustic signal generation and propagation, modulation and encoding, and reliable communication protocols. Moreover, techniques that leverage photoacoustic links in providing UAN services such as localization and topology management of UANs are being developed. This paper opts to provide an overview of such an area of research, summarizes progress made to date, and highlights open issues that are being or are to be worked on. To the best of the authors’ knowledge, there is no similar article in the literature. The hope is that scientists, practitioners, and developers could gain sufficient background, understand the unconventional design challenges, and know the state of the art on photoacoustic communication. The contributions of this paper can be summarized as: (1) a detailed summary of photoacoustic signal generation, especially how laser-induced breakdown occurs and consequently breakdown shockwave and bubble expansion-collapse shockwaves are produced, (2) elaborate discussion of different key properties of the linear and nonlinear photoacoustic effect, (3) an analysis of the challenges associated with the photoacoustic based communication, (4) a report on the current state of the photoacoustic communication research, (5) a highlight of the potential of photoacoustic enabled underwater localization, and (6) an outline of open research issues associated with the air-to-water photoacoustic communication. The paper is structured as follows. Section 2 describes how to generate photoacoustic signals. Section 3 and Section 4 discuss the properties of linear and nonlinear photoacoustic, respectively. Section 5 highlights the challenges associated with photoacoustic communication, and summarizes progress made to-date on developing the protocol stack and on enabling airborne-assisted underwater localization. Section 6 concludes the paper with a brief summary and an outline of open issues that warrants future work.

## 2. Photoacoustic Signal Generation

The photoacoustic (PA) effect is the formation of acoustic signals following the light absorption of the medium. For example, the PA signal can be generated when high intensity light impinges on transparent liquid, such as water. The laser-based PA transmitter uses a high-power laser to convert optical energy into acoustic energy at the air–water interface and provides a standoff method for transmitting acoustic signals from air into the water. Based on the energy density and irradiance imparted to the medium, this energy-transfer mechanism can be subdivided into a linear and a nonlinear domain. In the linear process, the light is exponentially attenuated by the medium, resulting in local temperature variations that give rise to volume expansion and contraction. These fluctuations in the density generate a propagating pressure wave. High intensity light, like laser pulses, only heats the medium and the physical state of the medium does not change in the linear PA process. Therefore, the generated acoustic power is proportional to the laser power [39]. On the other hand, in nonlinear PA the physical state of the medium changes during energy transfer, and the acoustic pressure is nonlinearly related to the laser power. The optical breakdown leads to plasma generation at locations where the breakdown threshold is exceeded in a nonlinear PA process. This plasma formation is associated with breakdown and cavitation bubble expansion–collapse shockwaves to generate acoustic signals. The generation of Linear PA is relatively simpler and heavily studied in medical research [40], but the nonlinear PA is a more intricate process. Therefore, details on the breakdown threshold, plasma size-shape, and shockwaves are discussed in the balance of this section.

### 2.1. Laser-Induced Breakdown

The laser-induced breakdown in transparent liquid (e.g., water) is a nonlinear process where a combination of photoionization, inverse Bremsstrahlung absorption, and cascade (avalanche) ionization produces free electrons in larger amounts and, together with the positive ions, form a plasma [41,42,43]. High-intensity laser pulses can create underwater plasma by focusing into a small spot so that the laser irradiance (*I*) surpasses the breakdown threshold irradiance. The laser irradiance can be calculated from the peak power of the laser divided by the focal spot area. The peak power of the laser is measured by the ratio of laser pulse energy (*E*) and pulse duration ($\tau_{L}$). Thus, the laser irradiance is [44],
(1)$$I=\frac{E/\tau_{L}}{A_{f}}$$

Here, the focal spot area, $A_{f\;}=\pi \omega_{0}^{2}$ where spot radius ($\omega_{0}$) can be calculated by,
(2)$$\omega_{0}=\frac{\lambda fM^{2}}{\pi \;\left(D/2\right)}$$

In Equation (2), f is the focal length of the lens, D is the diameter of the laser beam, λ is the wavelength of the laser beam, and $M^{2}$ is the beam propagation ratio. $M^{2}$ signifies how close a laser is to a single-mode TEM00 beam and defines how small a beam waist can be focused. When such beam propagation ratio, $M^{2}$, equals 1, it indicates the perfect Gaussian condition where the focused spot is diffraction limited. Thus, the diffraction-limited focus spot radius is,
(3)$$\omega_{0}=\frac{2\;\lambda }{\pi }.\frac{f}{D}$$

Based on Equation (3), in order to decrease the focal spot radius, a lens with a shorter focal length needs to be used or the laser beam diameter needs to be increased. Here, the ratio of focal length to beam diameter is known as *f*-number (f/#=f/D). *f*-number can be calculated by the focusing angle (θ),
(4)f/#=12×tanθ2

From Equations (3) and (4), we can observe that the *f*-number and the focus spot radius will increase if the focusing angle decreases. Similarly, the *f*-number and the focus spot radius will decrease if the focusing angle increases. The laser irradiance can be boosted to exceed the breakdown threshold by increasing the pulse energy or decreasing the *f*-number. The breakdown threshold is related to the laser parameters and the dependence of wavelength, pulse duration, and focusing geometry on breakdown threshold for plasma formation is investigated theoretically and experimentally in [41,42,43,44,45,46,47,48,49,50,51,52,53,54,55,56]. Table 1 shows the experimental breakdown thresholds for water for different pulse durations, wavelengths, and focusing angles with measured spot diameter.

From Table 1, we can observe that the irradiance threshold values are in the order of 10^11^ W/cm^2^ for a few nanosecond pulse durations and rise up to 10^13^ W/cm^2^ for 100 femtosecond pulse duration. The energy threshold for breakdown decreases by reducing the pulse duration, and the irradiance threshold increases. The irradiance required for optical breakdown for 30 ps laser pulses is on average 5.9 times higher than 6 ns laser pulses [55]. The optical breakdown threshold is slightly higher at 1064 nm than 532 nm. The breakdown threshold at 532 nm for 6-ns pulses is 0.57 times lower than the corresponding value at 1064 nm, while it is 0.83 times lower for 30-ps pulses [55]. Kennedy et al. [57] also observed similar results. All the spot sizes in Table 1 are measured in the experiment. The measured and diffraction limited spot size is nearly identical for small focusing angles. However, the measured spot size deviates from diffraction-limited spot size calculation for large focusing angles. Vogel et al. [55] did not observe any systematic dependence between breakdown threshold and spot size, which is in contrast to the work of Loertscher [48] and Docchio et al. [46]. Such contrast is because Loertscher used large focusing angles between 8° to 16° and assumed diffraction limited conditions to calculate $I_{th}$ while Docchio used very small focusing angle ≤1.7 where $I_{th}$ is influenced by self-focusing of the laser beam.

### 2.2. Plasma Size and Shape

The plasma formation is associated with breakdown shockwave, cavitation bubble expansion and collapse, thus creating acoustic signal. The generated acoustic signal depends on the plasma, and the size and shape of the plasma need to be controlled to control the acoustic signal. The plasma length ($z_{max}$) reached at maximum irradiance for a laser pulse with Gaussian shape and beam profile is [45],
(5)$$z_{max}=\frac{\pi \omega_{0}^{2}\;}{\lambda }\sqrt{\beta -1}$$
where, the normalized laser pulse energy, $\beta =\frac{E}{E_{th}}=\frac{I}{I_{th}}$ and $\frac{\pi \omega_{0}^{2}\;}{\lambda }$ is the Rayleigh range. The dependency of maximum plasma length ($z_{max}$) on the focusing angle (θ) is given in [55] as,
(6)$$z_{max}=\frac{\lambda }{\pi \tan^{2}\frac{\theta }{2}}\;\sqrt{\beta -1}$$

The dependence of $z_{max}$ on the laser wavelength, pulse energy, focusing spot size and angle is clearly visible in Equations (5) and (6); however, the dependence of $z_{max}$ on laser pulse duration is implicit. Since the breakdown threshold varies with pulse duration, as shown in Table 1, determining the normalized laser pulse energy requires knowledge of the breakdown threshold at the pulse duration of interest, which in essence affects $z_{max}$. In Figure 2, experimentally measured plasma lengths generated by 6 ns and 30 ps laser [55] are compared with the calculated values from Equation (6). For picosecond pulses, we can observe that the calculated plasma length is nearly identical to the experimentally measured values but is not as close for nanosecond pulses. At equal normalized laser pulse energy β, the nanosecond plasma is longer than the picosecond plasma. This is in contrast with Docchio’s [45] moving breakdown model where the plasma length is predicted to be independent of pulse duration. One reason for the relative deviation of nanosecond plasma can be the breakdown threshold which is influenced by UV radiation emitted by the plasma. The breakdown threshold decreases due to plasma radiation for nanosecond pulses. However, the picosecond breakdown process is less influenced by plasma radiation and remains approximately constant during breakdown. As a result, the plasma length generated by nanosecond laser pulse grows further than predicted using Equations (5) and (6), which assume a constant threshold. Optical aberration and diffraction-limited calculations are another cause for getting longer plasma length from experiments.

The dependence of plasma size and shape on laser pulse energy and the focusing geometry is investigated in [44,55,58,59,60,61,62]. In [55], the authors showed that the increase of plasma length with growth in laser pulse energy is more evident for picosecond pulses than for nanosecond pulses. For the same laser pulse energy, plasma generated by a picosecond laser is always longer than the nanosecond laser because of the lower breakdown threshold (see Table 1). However, nanosecond laser generated plasma lengths are shorter than picosecond laser generated plasma at equal normalized laser pulse energy (Figure 1). The focusing angle (θ) and focal spot radius ($\omega_{0}$) are inversely related and the plasma length is strongly dependent on the focusing angle and spot size. Increasing the focusing angle decreases the plasma length for a fixed laser pulse energy. A shorter plasma length implies a more spherical shape; the shape becomes more cylindrical as the plasma length elongates. Unlike the case of continuous plasma on solid targets, multiple plasmas can be formed at positions along the focal volume in water. Tian et al. [59] have summarized the reasons for multiple plasma formation and deduced that it is related to the focusing geometry of the laser beam. They demonstrated that the discrete and irregular plasma created in numerous locations could be transformed into continuous and stable plasma with a single core fixed at the laser focal point by increasing the laser focusing angle. Thus, the pulse-to-pulse repeatability is significantly improved with a factor ~2.7 and the plasma becomes more compact with a stronger emission with a factor ~3.0. Moreover, the double-lens combinations with shorter effective focal length can generate more condensed plasma, equating to a shorter plasma using the same laser energy [60]. As a result of this behavior, the moving-breakdown range can be effectively decreased while the total emission intensity of plasma is increased. In [61], it has been shown that spherical aberrations increase the breakdown threshold energy, resulting in the formation of multiple elongated plasmas with weak emissions. Using fast imaging and spectroscopic techniques, it has been demonstrated that a compact plasma with stronger emissions could be generated by aberration minimized focusing with larger focusing angles. Sinibaldi et al. [62] have studied the plasma sphericity index (*ζ*) as a function of laser pulse energy (E) and focusing angle (θ), where ζ(E,θ)=wzmax. The results, which are redrawn in Figure 3, demonstrate that the plasma is more spherical for higher focusing angles, and the sphericity index is around 0.7–0.8 at threshold energy. However, the sphericity index is capped by 0.4 at large energies, regardless of the focusing angle. A very small focusing angle means focusing with a high *f*-number generates filament-like plasma. For example, Vogel et al. observed filament formation at 1.8° focusing angle for the 6-ns pulses and at 1.7° focusing angle for the 30-ps pulses because of self-focusing [55]. They also reported filament formation at larger focusing angles (4° and 8.5°) with pulse energies far above the breakdown threshold.

### 2.3. Breakdown and Bubble Collapse Shockwaves

The laser-induced breakdown and bubble collapse produce shockwaves during the nonlinear PA process. First, the breakdown shockwave emission occurs during the plasma formation. Subsequently, the generated cavitation bubbles expand, collapse, and re-expand, creating additional in-water acoustic signals of various frequencies, depending on the bubbles’ size. In [63], the breakdown shockwave velocity ($u_{s}$) has been measured using time-resolved photography and then determined the shock pressure as,
(7)$$p_{s}=c_{1}p_{0}u_{s}\left(10^{\frac{u_{s}-c_{0}}{c_{2}}}-1\right)+p_{\infty }$$

Here, $\rho_{0}$ is the water density before being compressed by the shockwave, $c_{0}$ is the normal sound velocity in the water, $c_{1}=5190\;{\mathrm{ms}}^{-1}$, $c_{2}=25306\;{\mathrm{ms}}^{-1}$, and $\rho_{\infty }$ is the hydrostatic pressure. However, the determination of shock pressure is accurate only in a small region of less than one mm^3^ around the emission center, where the difference between shockwave and sound velocities can be easily detectable. Figure 4 shows experimentally measured shockwave velocities for nanosecond and picosecond pulse durations where we can observe the velocity is higher for nanosecond pulses than the picosecond pulses. Nanosecond pulses produce a higher electron density than picosecond pulses, which results in a higher plasma absorption coefficient, stronger plasma radiation, higher plasma temperature, and stronger mechanical effects such as shockwave emission and cavitation bubble expansion [55].

When the breakdown shockwave separates from the plasma, a volume of high pressure and temperature remains, which is out of equilibrium with the surroundings. Such volume expansion is slower than the breakdown shockwave release. Both work against the ambient static pressure ($p_{stat}$), which in underwater steps is a combination of the hydrostatic pressure due to the weight of the water column and the atmospheric pressure on the free surface. The expanding volume is then filled with water vapor, exerting an internal pressure $p_{v}$ and leading to the formation of a spherical bubble of maximum radius $R_{max}$. The energy stored in the cavitation bubble is directly proportional to the maximum bubble radius raised to the third power [63],
(8)$$E_{B}=\frac{4\pi }{3}\left(p_{stat}-p_{v}\right)R_{max}^{3}$$

The interior pressure is $p_{v}$ (2.33 kPa in water at 20 °C) is much smaller than the outer pressure $p_{stat}$ (typically about 100 kPa) after reaching the maximum radius of the bubble [64]. Then the bubble is compressed back to a small volume, and a shockwave generates upon the collapse of the bubble. The bubble expands again to a smaller maximum radius because of the energy lost by the previous bubble collapse shockwave and the rebound bubble is no longer spherical [64]. The maximum bubble radius is related to its oscillation period $T_{B}$ which is twice the collapse time ($T_{c}$) by Rayleigh equation [65],
(9)$$R_{max}=T_{B}/2\times 0.915\;{\left(\frac{\rho_{0}}{\rho_{\infty }-\rho_{v}}\right)}^{0.5}$$

Figure 5 illustrates the breakdown shockwave and bubble expansion and collapse shockwaves generated by our experiment with 1064 nm Nd:YAG laser with 6 ns pulse duration. We can observe that each successive shockwave has less peak pressure because the previous shockwave containing mechanical energy is already emitted, and heat is exchanged with the surrounding environment. Therefore, the remaining mechanical energy is reduced because no new energy is imparted from the laser source during this time. Thus, the breakdown shockwave has higher peak pressure than bubble expansion and collapse shockwaves. The number of acoustic transients generated by the cavitation bubble is related to the total mechanical energy made available by the laser pulse energy. Here, three transients are generated by bubble collapse before the energy needed for additional transients is dissipated. The generated acoustic signal is also dependent on the focusing geometry. Because short and highly absorbing plasmas can be generated at large focusing angles, which allows a well-localized energy deposition at a low breakdown threshold. Large focusing angles are associated with a high conversion efficiency into mechanical energy and thus a large potential to induce higher acoustic pressure. However, the mechanical effects can be significantly reduced by decreasing the laser pulse duration [54].

In this section, we have summarized prior research work related to linear and nonlinear PA signal generation. In the next two sections, we describe some properties of linear and nonlinear PA.

## 3. Properties of Linear Photoacoustic

When high energy light is absorbed by the water, depending on the energy density and irradiance of the light, the physical state of the water medium either changes or does not change. If the property of the water medium does not change, the optoacoustic process is called linear. In linear optoacoustic, absorption of high light energy creates local temperature fluctuation in the water medium which causes volume fluctuation. Eventually this volume fluctuation creates a propagating acoustic signal. The linear categorization is because the power of the generated acoustic signal is proportional to the input laser power. In contrast, when irradiance of light energy is higher than a certain threshold level, the property of the water medium changes, and the optoacoustic process is called nonlinear. As the name suggests, in nonlinear optoacoustic power of the generated acoustic signal is not proportional to the input laser power. This section focuses on the linear optoacoustic mechanism. Achievable communication ranges, laser modulation configurations, and data rates are addressed. Usually, the generated acoustic signal by the optoacoustic effect is very broadband. Yet, when an acoustic wave travels a long distance in the water medium, the high frequency components attenuate faster which eventually causes a very weak signal detection at the receiver side. Hence, we will start our discussion by frequency domain analysis and how to generate narrow-band acoustic signals.

### 3.1. Frequency Domain Analysis

Figure 6 shows a general geometry of the optoacoustic mechanism where a laser beam with radius a has a normal incidence in the water medium. The pressure spectral response at point P which is R distance away from the source can be expressed as [66]:(10)$$P\left(R,\omega \right)=A\left(\frac{x^{2}}{x^{2}+1}\right)e^{-{\left(\frac{Sx}{2}\right)}^{2}}\frac{\sin\left(\frac{N\delta x}{2}\right)}{\sin\left(\frac{\delta x}{2}\right)}$$
where,
(11)$$A=-\left(\frac{\beta P_{o}}{2\sqrt{\pi }c_{p}}\right)\frac{e^{ikR}}{R}\left(\frac{\tau }{\tau_{\mu }}\right)e^{-\frac{\omega^{2}\tau^{2}}{4}}e^{i\frac{\left(N-1\right)\omega T}{2}}$$

Here, x is referred to as the angular frequency which is normalized by a time constant $\tau_{\mu }$, where $\tau_{\mu }=\frac{\cos\theta }{\mu c}$. In Equation (10), S is defined as the “slenderness” of the optoacoustic source and calculated as $S=\frac{\tau_{a}}{\tau_{\mu }}$, where $\tau_{a}=\frac{a\sin\theta }{c}$ is the characteristics transit time across the laser beam diameter. δ is the normalized inverse repetition rate of the laser source and is defined as $\delta =\frac{T}{\tau_{\mu }}$. All the other parameters in Equations (10) and (11) are defined in Abbreviations. Equation (10) shows the spectral response of the generated acoustic signal for various laser parameters such as laser beam diameter, a, laser pulse repetition rate ($\frac{1}{T}$), laser power, $P_{o}$ and some environmental parameters such as optical absorption coefficient of water, μ observation angle, θ etc. Generally, this spectral response is very broadband. Yet, by choosing proper laser repetition rate and observation angle, narrow band signal generation is also possible.

### 3.2. Laser Repetition Rate

Choosing an arbitrary laser repetition rate creates a very broadband signal. By carefully choosing the value of repetition rate a narrowband signal generation is possible [66]. In order to do that, an optimum value of δ is required to maximizes P(R,ω) because δ is directly related to the laser repetition rate. To better understand the effect of δ on the acoustic power spectrum we can rewrite Equation (10) as:(12)P(R,ω)=A.F(x).G(x)
where,
(13)$$F\left(x\right)=\left(\frac{x^{2}}{x^{2}+1}\right)e^{-{\left(\frac{Sx}{2}\right)}^{2}}$$
and
(14)$$G\left(x\right)=\frac{1}{N}\left|\frac{\sin\left(\frac{N\delta x}{2}\right)}{\sin\left(\frac{\delta x}{2}\right)}\right|$$

Figure 7 shows the effect of δ on F and G. Since power spectral response P(R,ω) is the product of F and G, P will be maximum when both F and G are high. If one of them is low the products become low. In Figure 7a, when the value of δ=6, we observe that multiple peak values of *G* overlap with a high value of *F*. However, in Figure 7b, when δ=2.4 only one peak value of *G* overlaps with a high value of *F*, which indicates that having δ=2.4 creates a narrowband signal.

### 3.3. Observation Angle

Another way to generate a narrowband linear optoacoustic signal is by properly choosing the incident angle of the laser light and the observation angle of the receiver [39,67]. A wavy water surface imposes further complexity to determine these angle values. Figure 8 shows the relationship between these angles for a wavy water surface. With the help of Equations (1)–(12) in [67], the power spectral density can be rewritten as:(15)$$P\left(r,\omega \right)=A_{0}\left(\mu ,p_{0},\omega ,r,\tau ,T\right)\times D\left(\theta_{r},\theta ,\varphi ,\omega ,\mu ,a\right)$$
where, $A_{0}$ is the amplitude term whose value mainly depends on the laser parameters, $\mu ,p_{0},\omega ,r,\tau ,\;\mathrm{and}\;T$, and *D* is called directionality factor whose value depends on environmental parameters and observation angles, $\theta_{r},\theta ,\;\mathrm{and}\;\varphi$. Through some rigorous theoretical analysis, it is shown in [67] that when the sum of refracted angle, $\theta_{r}$ and vertical observation angle, θ is close to $90^{{}^{\circ}}$, narrowband acoustic signal generation is possible. Figure 9a,b illustrate these two scenarios where a laser beam is transmitted at an angle such that the sum of θ and $\theta_{r}$ is around $90^{{}^{\circ}}$. Figure 9a,b are based on a flat-water surface. Figure 9c shows the case for a wavy water surface. In [67], it was concluded that if the height of the water wave is small enough, a change in θ and $\theta_{r}$ does not impact their sum, i.e., if $\theta_{r}$ decreases θ increases in a similar amount so that their sum remains 90°.

From the above discussion we can conclude that, by choosing the optimum repletion rate and incident angle of the laser beam, a narrow band acoustic signal generation is possible for a linear optoacoustic.

## 4. Properties of Nonlinear Photoacoustic

In this section, we discuss the generation of broadband acoustic signals by a short duration single laser pulse and how to control the spacing between the frequency components of the spectrum, the acoustic source level, acoustic pulse duration, and the propagation directivity.

### 4.1. Broadband Signal

Underwater acoustic signals can be generated from a remote, aerial location by using a high-energy pulsed laser. The nonlinear PA signal generation is more efficient, i.e., greater source level but less controlled approach for producing underwater acoustic signals than the linear mechanism. We have experimentally investigated the nonlinear PA signal by a 1064 nm, 6 ns pulse duration Q-switch Nd:YAG laser. Underwater acoustic signal is generated by the laser focusing with a 7.5 cm lens in water and plasma is created approximately 2 cm below from the water surface. A hydrophone with frequency range 15 Hz to 480 kHz is placed at 0° angle relative to the norm at the incident point of the laser beam to monitor the generated acoustic signal. Figure 10a illustrates the acoustic signal generated from a single laser pulse. After laser pulsing, little flashes of white light were noticed in the water, which is connected with the plasma generated during each optical breakdown event. The frequency spectrum of the generated acoustic signal is shown in Figure 10b. We can observe that a single laser pulse produces a broadband acoustic signal with considerable acoustic energy. The frequency components of bubble oscillation, particularly the delay between the breakdown and bubble shock waves, resulting in a variation of the frequency content, complicating the acoustic spectrum.

The generated acoustic signal and its associated spectral character depend on the laser parameters, and thus can be controlled by the proper choice of the laser modulation parameters, i.e., laser wavelength, pulse energy, repetition rate, and laser beam focusing. Blackmon et al. [68] investigated and demonstrated a means of deterministically controlling the spectrum of the underwater acoustic signal by varying the laser pulse repetition rate. The following equation gives the general time domain expression for the pressure waveform as a function of range and vertical observation angle,
(16)$$\begin{array}{cc} p\left(r,\theta_{v},\;t\right)= & P_{m}\left(r,\theta_{v}\right)\overset{N-1}{\underset{n=0}{\sum }}\exp\left[-\frac{\left(t-nT_{R}\right)}{\tau \left(r\right)}\right]u\left(t-nT_{R}\right)\\ & +\underset{j}{\sum }P_{Bj}\left(r,\theta_{v}\right)\overset{N-1}{\underset{n=0}{\sum }}\exp\left[-\frac{\left(t-T_{Bj}-nT_{R}\right)}{\tau_{Bj}\left(r\right)}\right]\times u\left(t-T_{Bj}-nT_{R}\right) \end{array}$$
where,
N= number of laser pulses,$T_{R}=$ laser pulse repetition period,$T_{Bj}=$ time delay between peak of plasma generated transient and peak of *jth* cavitation generated transient,$P_{m}\left(r,\theta_{v}\right)=$ peak pressure of plasma generated acoustic transient as a function of range,$P_{Bj}\left(r,\theta_{v}\right)=$ peak pressure of the *jth* cavitation generated acoustic transient as a function of range,τ(r)= time constant of plasma-generated acoustic transient as a function of range,$\tau_{Bj}\left(r\right)=$ time constant of the *jth* cavitation generated acoustic transient as a function of range.

In Equation (16), the pressure waveform contains an optical breakdown pressure term followed by a summation of time-delayed, cavitation bubble generated pressure terms; each pressure term is a scaled exponential acoustic transient. The corresponding acoustic pressure spectrum’s magnitude is given as [68],
(17)$$\left|P\left(r,\theta_{v},\;\omega \right)\right|=\left|\frac{\sin\left(\frac{N\omega T_{R}}{2}\right)}{\sin\left(\frac{\omega T_{R}}{2}\right)}\right|\;\left|\left[\frac{P_{m}\left(r,\theta_{v}\right)\tau \left(r\right)}{1+j\omega \tau \left(r\right)}+\underset{j}{\sum }\frac{P_{Bj}\left(r,\theta_{v}\right)\tau_{Bj}\left(r\right)}{1+j\omega \tau_{Bj}\left(r\right)}\;\exp\left(-j\omega T_{Bj}\right)\right]\right|$$

Here the pressure spectrum magnitude is given as the product of the single-pulse Fourier transform magnitude (specified in the bracketed term) and the periodic Sinc function or Dirichlet function magnitude with maxima at the laser repetition frequency and its harmonics. Blackmon and Antonelli demonstrated that the separation between oscillations in the spectrum correlated with the laser repetition rate [68]. By controlling the frequency placement and spacing of components in the spectrum by the generation of acoustic transients associated with controlled periodic laser pulsation, the laser’s pulse repetition rate can be used to broadcast specified acoustic frequencies. However, due to the transient nature of the nonlinear optoacoustic conversion process, the overall spectrum remains relatively broadband and constant throughout.

### 4.2. Remote Underwater Acoustic Source

In order to generate a remote underwater laser acoustic source, the Naval Research Laboratory (NRL) investigated the physics of intense underwater laser propagation and acoustic generation, including linear group velocity dispersion (GVD), nonlinear refractive index effects such as nonlinear self-focusing (NSF), filamentation, self-phase modulation; scattering, absorption, and laser-induced breakdown. An ultrashort laser pulse can propagate relatively long distances underwater (up to distances on the order of the attenuation length, approximately 10 m in seawater) at a moderate intensity using the NSF and GVD mechanisms. Then the laser pulse quickly converges to an intense focus within a few centimeters at a predetermined remote location and generates a plasma and an acoustic shock wave at this location [69]. Due to GVD the laser optical pulse longitudinally compresses and increases the intensity of the negatively chirped optical pulse, triggering the self-focusing effect for transverse compression. The propagation distance $L_{GVD}$ needed to produce maximum longitudinal pulse compression for a negatively chirped pulse in a medium with linear GVD is approximately [69],
(18)$$L_{GVD}\approx \;\frac{T\left(0\right)}{\beta_{2}\partial \omega }$$

Here, T(0) is the initial pulse duration, ∂ω is the frequency bandwidth and $\beta_{2}$ is the GVD parameter. Thus, the longitudinal compression range can be controlled by controlling the initial pulse length and/or the laser bandwidth. Longitudinal pulse compression increases the intensity of the negatively chirped optical pulse and transverse compression of the pulse generally occurs when the intensity of the pulse exceeds a threshold represented by,
(19)$$P_{critical}=\frac{\lambda^{2}}{2\pi n_{0}n_{2}}$$
where $n_{0}$ is the linear index of refraction and $n_{2}$ is the nonlinear index of refraction. $P_{critical}$ is of the order of 1 MW for visible wavelengths in water. A characteristic distance for the transverse compression by NSF is approximately [70],
(20)$$L_{NSF}=\frac{z_{R}}{\sqrt{\frac{P\left(z\right)}{P_{NSF}}}-1}$$

Here, $z_{R}$ is the Raleigh range. For optimal pulse compression, initial beam power and beam radius should be selected such that $L_{NSF}=L_{GVD}$ and the transverse and longitudinal compression occur simultaneously at a chosen distance. Thus, the intensity of the pulse is high enough to cause optical breakdown at a chosen distance and generate the acoustic signal. The laser wavelength is preferably chosen to have low attenuation in water, as attenuation is highly dependent on the wavelength. An attenuation length $\left(L_{atten}\right)$ can characterize attenuation of light in water, with the beam intensity decreasing with propagation distance z according to $I\left(z\right)=I\left(0\right)\exp\left(-z/L_{atten}\right)$. The underwater propagation path length should be selected to be less than $L_{atten}$ for applications where maximum energy is required at the acoustic source. However, the total underwater propagation path can be a few times greater than the attenuation length for lower energy applications. The maximum transmission (and least absorption) occurs in pure water in the wavelength range 300–500 nanometers, with a maximum attenuation length of around 50 m in this region. However, the attenuation length in seawater is typically 5 to 10 m because of impurity concentrations. The global average attenuation length is approximately 4 m but can reach 10 m or more in relatively clear ocean water [70].

NRL has investigated underwater optical filaments, including the determination of optimal generation parameters, maximum propagation length, and filament plasma lifetime to create the underwater acoustic source. Filamentation is observed in water when laser power is above the critical power ($P/P_{critical}>1$) with focused above a critical *f*-number. Underwater filamentation is observed with $P/P_{critical}\approx 2$, *f*-number = 80 for nanosecond lasers where filaments propagated up to 25 cm (Rayleigh length 2 cm) [71]. However, both nanosecond and femtosecond pulses cause longitudinally localized laser absorption and generate a single acoustic shock at small *f*-number [71]. Underwater optical filaments over 55 cm (Rayleigh length more than 35 cm) generation in water with nanosecond pulse duration laser is reported in [72], where the filament radius (50±10 μm) is much larger than the few-micron diameter generated with the femtosecond laser. However, the underwater filament sizes and propagation characteristics of ultra-short and nanosecond pulses may differ due to a variety of factors, including significantly different light intensity in the filament, distinct ionization mechanisms and rates, and effects unique to the large bandwidth of the ultra-short laser pulses. The advantage of using longer laser pulses like nanosecond pulses is that they contain more energy, resulting in high conductivity plasma.

### 4.3. Acoustic Pulse Duration and Dominant Frequency

The generated acoustic pulse duration is dependent on plasma volume where the higher laser pulse energy and larger plasma volume produce longer acoustic pulses. However, the acoustic pulse length is not necessarily the same in all acoustic propagation directions. A laser pulse can produce a disc-shaped ionized volume from only GVD-induced longitudinal compression to generate breakdown [70]. This can produce longer acoustic pulse lengths in directions parallel to the plane of the disc. Alternatively, for applications that require short underwater propagation distances, optical pulses with little or no frequency chirp can be created that rely solely on NSF effects to achieve laser-induced breakdown intensities [70]. Milián et al. [73] numerically and experimentally showed that the nonlinear energy deposition and plasma density distribution depend on the input pulse chirp and focusing conditions. First, they demonstrated that the lowest optical transmission corresponds to negatively pre-chirped pulses that form a plasma with the maximum possible energy. Then further negative pre-chirping of the laser pulse results in a plasma volume with maximized length where the electron density is still relatively high. Finally, plasma densities are maximized at the expense of plasma channel length for even greater negative values of the input chirp.

The propagation of acoustic pulses is dependent on the overall distribution of their component frequencies, specifically on the frequency with the greatest power within that distribution, represented by $f_{peak}$. A smaller $f_{peak}$ corresponds to a longer range because the attenuation of a propagating acoustic pulse is proportional to the square of $f_{peak}$. A way to reduce $f_{peak}$ by more than an order of magnitude (from 200 kHz to 15 kHz) is developed by NRL [74]. The key to low-frequency acoustic generation is shaping the underwater plasma volume, where more elongated volumes produce longer-duration acoustic pulses containing more energy at low frequencies. T.G. Jones et al. proposed a technique to generate meter-scale elongated plasma utilizing two laser pulses with energy spectral density (ESD) centered near 1 kHz [75]. In this technique, a first laser pulse is fired into the water to form an underwater optical filament coinciding with a low energy plasma. The second laser pulse is used to heat the filament plasma, causing the formation of an extended superheated plasma volume. The two laser pulses can be delivered simultaneously or sequentially, with the second pulse following up to the filament plasma lifetime after the first. The two-laser pulse technique enables the generation of much longer energetic plasmas, of order 1 m in length or longer, than the plasmas generated with a single pulse, which have a length up to order 5 cm [75].

The acoustic pulse duration ($\tau_{ac}$) in any direction of acoustic propagation is dependent on the plasma dimension (d) in that direction and the initial shock propagation speed (${\vec{V}}_{shock}$) [75],
(21)$$\tau_{ac}\approx \;\frac{d}{{\vec{V}}_{shock}}$$

Therefore, spherical plasma can generate isotropic acoustic pulse duration, and only increasing laser pulse energy is an inefficient means to increase the pulse duration because $\tau_{ac}\sim E^{1/3}$. On the other hand, elongated plasmas produce anisotropic pulse duration even though ${\vec{V}}_{shock}$ is similar in all directions. This is because the elongated plasma has a shorter dimension $d_{⊥}$ in a direction normal to the laser beam axis than the dimension $d_{\|}$ in a direction parallel to the beam. Thus, for elongated plasma $\tau_{\|}>\tau_{⊥}$.

### 4.4. Acoustic Source Level and Directionality

The nonlinear PA process can generate intense acoustic signals. The PA generation efficiency using ionization (≤10%) is higher than the efficiency (≪0.1%) for thermal expansion. Figure 11 shows the acoustic source level (SL) as a function of pulse energy for narrowband and broadband lasers based on data collected by NRL [74,76,77]. The SL is measured at 1 m distance from the acoustic source with 10^−6^ Pa reference pressure. We can observe the SL increases with laser pulse energy which is expected but for laser pulse energy less than 10 mJ, narrowband nanosecond laser generates a higher SL than broadband femtosecond laser for approximately the same pulse energy. Nanosecond pulses create higher electron density which in turn results in a higher absorption coefficient of the plasma and stronger acoustic signal. Noak et al. [78] also demonstrated that mechanical effects are significantly reduced for shorter laser pulses because the energy required to produce optical breakdown decreases with decreasing pulse duration. Thus, a more significant fraction of energy is required to overcome the heat of vaporization with shorter laser pulses. Yellaiah and Kiran [79] investigated the input pulse duration effects on the laser induced underwater acoustic signals with nanosecond and picosecond pulse duration, where they found nanosecond pulses achieve better energy conversion efficiency. Shi et al. [80] studied the acoustic signal strength by varying the distance between the focusing lens and the water surface and found that the strength of the acoustic signal decreases as the distance between the focusing lens and the water surface decreases due to greater optical refraction.

The directionality of the generated acoustic signal is also dependent on the plasma shape. The generated acoustic signal has peak pressure in the direction of the laser beam for femtosecond laser pulses where a more localized, spherical plasma volume is observed just below the water surface [71]. Here the femtosecond breakdown volume is axially localized due to short pulse length and GVD-induced pulse stretching. However, the peak pressure is perpendicular to the laser propagation direction for nanosecond laser pulses, where more elongated plasma is observed well below the water surface [71]. Figure 12 illustrates the generated SL with observation angle relative to laser propagation direction for different laser parameters; the data was collected by NRL [71,74,77]. As indicated by the plots, the SL increases with the observation angle for narrowband nanosecond laser pulse and decreases with the observation angle for broadband picosecond laser pulse.

## 5. PA-Based Communication and Localization

The photoacoustic mechanism is commonly used in medical imaging. Yet, it has a great potential as a means of air to underwater communication. In the previous sections we have discussed the generation and properties of linear and nonlinear photoacoustic signals. In this section, we will highlight some notable research work that has focused on exploring the photoacoustic effect in communication. We will start with discussion of some key challenges that are associated with photoacoustic communication.

### 5.1. PA Communication Challenges

The nonlinear PA is considered the most efficient method to generate acoustic signals in an underwater environment [68]. However, using the nonlinear PA causes generation of vapor clouds in the underwater environment due to the heating effect. These vapor clouds prohibit further generation of acoustic signals at the same location until the cloud decimates in the water [81]. Such vapor cloud decimation poses a theoretical bound on the speed of generating acoustic signal when nonlinear PA techniques are used and produces an upper bound on the achievable communication bitrate. To evade the vapor cloud, Blackmon et al. [68] suggest using a scanning laser that focuses the laser beam at different points in a region to avoid possible presence of vapor clouds.

Another key issue is that the photoacoustic effect is hybrid by nature. While a high energy light signal is used to transmit encoded data from the transmitter side, at the receiver side, an acoustic signal is received. This hybrid nature makes it very difficult to come up with a suitable modulation/demodulation or encoding/decoding scheme. Moreover, the generated acoustic signal in a PA process is highly directional, as we have discussed in Section 3. One needs to factor such directionality to make a successful communication. By controlling the shape of the plasma, the directionality of the acoustic beam may be controlled. Y. Brelet, et al., [82] have explored the PA effect and studied the shape of the generated underwater acoustic beam. They have found that the directivity of acoustic signal is concentrated on the plane perpendicular to the laser transmission direction.

### 5.2. Current State of PA Communication Research

In the previous subsection, we have discussed the challenges associated with the PA signals. In this section, we summarize some notable existing research work that opts to tackle these challenges. We will focus on different modulation schemes and some real communications that have been made from air to underwater using PA effect. Figure 13 shows the general block diagram of air to underwater communication using PA signal. A high-power laser light source is required for PA communication. Typically, a Q switch Nd: YAG laser is used for this kind of application [39]. Then a modulator is used to encode the information that needs to be transmitted. Additional mirrors and lenses might be required to focus the laser light inside the water. Since the generated acoustic signal is very broadband, a hydrophone with high bandwidth efficiency is required at the receiver end. The received signal is then amplified by an amplifier before being decoded.

Air to underwater PA communication using the general block diagram in Figure 13 has been the subject of quite a few research studies. In [39] Blackmon implemented different kinds of modulation techniques for linear PA. A mechanical chopper was used to modulate a laser beam at 10 kHz speed. An electro-optic Pockels cell modulator was used to generates different types of modulated signals such as sinusoidal modulation at 10 kHz, 4-bit BPSK modulation, 11 (5 to 15 kHz with 1 kHz spacing), 5 (8 to 12 kHz with 1 kHz spacing), and 2 (10 and 12 kHz) tone MFSK (multi-frequency-shift keyed) modulation, and FSK (frequency-shift keyed) modulation. Table 2 shows the experimental results of acoustic source level (SL) and achievable communication range for such modulation techniques. Here, communication range means in-water communication range.

The energy conversion rate for a linear PA process is very low. In order to improve the energy conversion rate, we need to maximize optical energy absorption by the water medium where the light energy is focused in the water medium. This could be done by introducing a passive relay in the water medium where light energy is focused. In [83] Z. Ji et al. used a low-cost passive relay in the air–water interface which improves the energy conversion loss. The cost of those passive relays is as low as US $1. The laser energy loss can reduce up to six orders of magnitude (25 J vs. 27 μJ). The intensity of the generated photoacoustic signal is proportional to the absorption coefficient of the light energy absorbing medium. A contrast agent (passive relay) with high optical absorption coefficient can significantly improve the light energy absorption in the water medium and thus minimizes the need of high laser energy.

The energy conversion can be further improved by introducing a nonlinear PA effect. In a nonlinear PA process, when the light energy exceeds a certain threshold level, plasma is also formed and not just the water medium is heated. The plasma can create a stronger acoustic signal than when pursuing a linear PA process. The intensity and directionality of the acoustic signal depends on the plasma shape. In Section 4, we introduced how to generate different shapes of plasma. In this section, we discuss how a modulation technique can be implemented by varying the plasma shape. The plasma shape generated by the nonlinear PA process, can be controlled either by varying laser pulse energy or focusing angle of the lens that is used to focus the laser light inside the water medium. In our previous work we proposed a novel modulation technique by varying the focusing angle of the lens which we named as Optical Focusing-based Adaptive Modulation (OFAM) [40]. The laser induced plasma can be considered as an antenna for the acoustic emission whose shape can be controlled by a varying focusing angle of the laser light. Typically, a shorter plasma is considered as an antenna whose directionality is spherical in shape. On the other hand, an elongated plasma is considered as a non-spherical or highly directional acoustic source. The idea of OFAM is to dynamically control the focal length by using electrically focus-tunable lens. Advanced lenses are available where the focal length of the lens is a function of electrical current [84]. Figure 14 shows how we can generate such different shapes of plasma using focus-tunable lenses and a lens driver. Through rigorous simulations and experiments, we have shown that OFAM can achieve better bit error rate (BER) performance.

The achievable bit rate using PA mechanism is very low (couple of Hz to kHz). So, sending large data like speech is not possible using PA mechanism. An interesting research work has been done by H. Jiang et al., where a speech signal is transmitted using a PA mechanism [85]. A speech recognition method has been proposed where speech is converted to characters and then encoded by the pulsed laser. The encoding process is done by varying the laser pulse repetition rate. Through experiments, they have shown that converting speech into characters can be reduced the required data rate thus PA mechanism can apply to send speech data.

Most of the above works have been done in the lab environment but NRL has successfully done open water demonstration of laser acoustic generation and propagation in 2010 and 2011 at the Lake Glendora Test Facility of Naval Surface Warfare Center, Crane Division [70]. The generated acoustic SL was about 190 dB, and boat-mounted hydrophones measured acoustic pulse propagation at distances up to 300 m (approximately 1000 ft). These field experiments confirmed that there is no significant ultrasonic attenuation at these ranges during acoustic propagation.

### 5.3. PA-Enabled Underwater Localization

Node localization is very important for many civil and military applications of underwater networks, particularly knowing where the nodes are, which allows correlation of the collected data and enables efficient management of the network topology. Localization strives to establish a coordinate system, which can be categorized as local or global. A local coordinate system reflects the position of nodes relative to one another. Such relative positioning could be realized through acoustic signals and may suffice for topology management. However, for correlating data, a global coordinate system would be needed, for which the underwater node location should be based on a terrestrial positioning system such as a GPS. Due to the major absorption loss of the higher frequency radio signals, it is not possible for an underwater node to rely on GPS receivers to infer its position. Global localization is possible by using a floating surface node or gateway which contains both radio and acoustic transceiver. Such a surface node or gateway receives the GPS coordinates from the satellite and then transmits them to underwater nodes over acoustic links. The transmission of GPS coordinates using acoustic signals can be divided in two categories based on the underwater node distance from the water surface: short-range and long-range localization.

Although the focus of this paper is on cross-medium communication using optoacoustic effect, a brief overview of underwater localization techniques is provided for completeness. Short-range underwater localization has been extensively studied and can be categorized into two main parts, namely, range-based and range-free localization [86]. In a range-based localization, underwater nodes need to leverage a ranging technique to obtain a relative location to several reference nodes. When global coordinates are required, surface buoys that can obtain their location using GPS signals are used to aid underwater nodes to localize themselves. Angle of Arrival (AoA), Received Signal Strength Indicator (RSSI), and Time of Arrival (TOA) and Time Difference of Arrival (TDoA) are the most-commonly used ranging techniques [29,86,87,88]. On the other hand, range-free localization techniques leverage the network topology and a beacon signal transmitted by a reference node to infer the locations of unknown underwater nodes [89]. However, range-based schemes usually perform better than range-free localization techniques [90], and hence most recent investigations target the range-based schemes. On the other hand, long-range underwater localization is often more challenging since underwater acoustic waves travel in an inhomogeneous environment with varying speeds. The varying speed of propagating acoustic signals results in different propagation behavior from the case of free space. Thus, modeling the acoustic communication channel in underwater configuration has been the target of multiple studies where the effect of Sound Speed Profile (SSP) onto the propagating acoustic signal is addressed [91,92,93,94,95,96]. In general, factoring in the SSP variability in underwater environments yields better estimates of acoustic propagation paths and eventually improves localization accuracy [97]. Moreover, a recent work is targeting to localize a swarm of underwater autonomous vehicles [98].

All the methods mentioned above require a floating surface node or gateway which constitutes a significant shortcoming. The deployment of such floating gateways is logistically complicated and constitutes security risk for the underwater networks. Several studies have been made to localize underwater nodes with the absence of a surface node where some means of communications are used to cross the air–water barrier. To cross the air–water barrier, a proposed technique relies on the use of Visual Light Communication (VLC) to transmit information from an off-water node to the underwater nodes [99,100]. Moreover, preliminary investigation has been made on how to use the VLC to localize underwater nodes using RSSI [101]. In the proposed scheme, the airborne node transmits its GPS location using the VLC communication towards the water surface where the underwater nodes reside. The underwater node then detects the VLC and determines its location based on the intensity of the received light signal. By factoring the loss encountered by the visual light through its path, the authors propose a localization scheme where the underwater node can know its relative location from the transmitter. Magnetic Induction (MI) is also used to overcome the challenges faced in the air–water interface. Several studies have been made on the use of MI to communicate in underwater environments [102,103].

Neither VLC nor MI can reach a longer distance in the underwater environment. Utilizing the PA effect to establish long distance cross-medium communication is deemed to be a more efficient method for underwater node localization. Particularly, an airborne unit generates virtual surface anchors to localize underwater networks. Leveraging a laser source, the airborne unit chooses a point on the water surface where a high energy pulsed laser is directed. Upon striking the water surface, the energy in the laser is absorbed by the water and acoustic signals are generated in underwater environments through the PA effect. By employing a suitable PA modulation methodology to transmit the GPS position of the airborne unit, Mahmud et al. [104] have devised a localization scheme that handles various configurations. Specifically, the airborne unit generates multiple virtual anchors that are non-collinearly positioned to localize underwater nodes. The authors also explore the ability to localize a mobile under node using a single airborne unit where multiple readings can be obtained by moving the underwater node from the range of one virtual anchor to another. Generally, research on PA-enabled underwater localization is in early stages and more progress is expected in the next few years.

## 6. Conclusions and Future Work

In this article, we have provided an overview of the state-of-the-art techniques for enabling air-to-underwater communication using the photoacoustic mechanism. We have described the generation, propagation, and characteristics of the acoustic signal and how to control the properties of such a signal through the laser beam. We have also outlined the research challenges in establishing photoacoustic communication links and reported progress made to date. The use of photoacoustic as a cross-medium communication has high potential to morph the classical way of approaching a solution in underwater applications. Since developments are still in an early stage, there are many open research issues that warrant further investigation. These issues include the following:Improving the bitrate is indeed warranted for photoacoustic links. To date the achievable bitrate is in the kHz range, which is extremely low for real-time audio and video communications. The use of traditional modulation techniques does not fully exploit the optoacoustic capabilities and often leads to slower data transmission rates. There is a clear need for innovative modulation and channel coding techniques that can leverage the different characteristics of the optoacoustic channel to increase the bitrate.Optimizing the laser source energy is an obvious means for strengthening the generated acoustic signal and improving the signal-to-noise ratio. The generation of a vapor cloud associated with each optoacoustic effect hinders the generation of new acoustic signals at the same location until that cloud decimates. Since high energy laser pulses tend to make larger vapor clouds that eventually require more time to decimate into the underwater environment, research is needed to find the optimum balance between the laser energy and the bitrate.A complete protocol stack is yet to be developed by taking advantage of the photoacoustic effect, not only the physical layer but also at the link and transport layers.Focusing laser light inside the water medium is extremely difficult in practice. In lab setups, one can do so by using focusing lenses. However, such an approach is not practical in an open ocean environment. Auto-focusing laser light could be a viable option that is worth investigation.Improving the bit error rate warrants more research efforts. Since the optoacoustic effect produces acoustic signals that propagate in an underwater environment, the channel parameters need to be estimated to ensure successful delivery of the transmitted data. Channel estimators for free space communication have been addressed in many recent publications [105,106]; yet little progress is made for underwater environments. Ideas portrayed in free space can be mapped to an underwater medium where the underwater dynamics are considered such as the medium inhomogeneity [107]. In the context of photoacoustic communication, underwater channel estimators can be leveraged for improving reception where the directionality of the acoustic beam can be controlled by the laser focusing angle. Furthermore, the creation of multiple bubbles in the nonlinear photoacoustic case can be exploited to mitigate the multipath effect where reflected waves either from the surface or bottom are used to establish communication and reduce the multipath fading [108]. Machine learning techniques would be a viable option for conducting such optimization, as demonstrated by recent work [106].Field testing of photoacoustic communication is necessary for gauging viability and performance. PA is an emerging area of research; most of the developments have been validated in lab settings using water tanks. The exception is work done at the US naval research laboratory where some sea-based experiments were conducted. As noted in Section 5.2, these experiments mainly focused on physics to explore the properties of the photoacoustic effect. Hence, more effort should be made to open water settings.

## Figures and Tables

**Figure 1 sensors-22-04224-f001:**
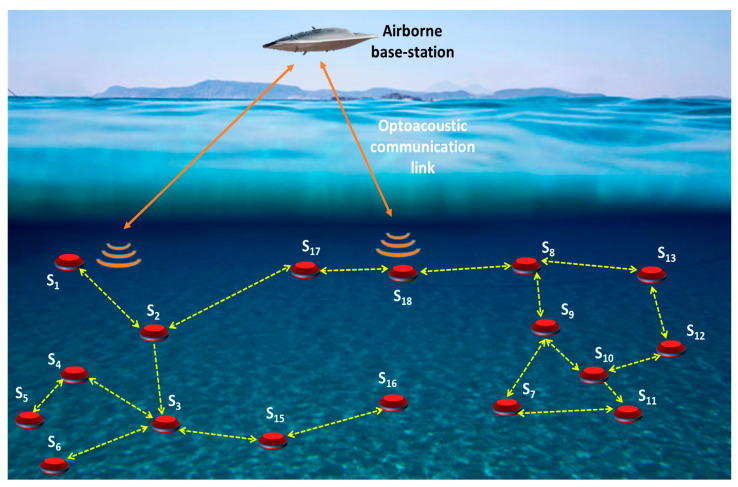
Underwater networking scenario showing nodes utilizing acoustic links to establish data routes to multiple destinations. The network is interfaced to an aerial vehicle through optoacoustic communication links.

**Figure 2 sensors-22-04224-f002:**
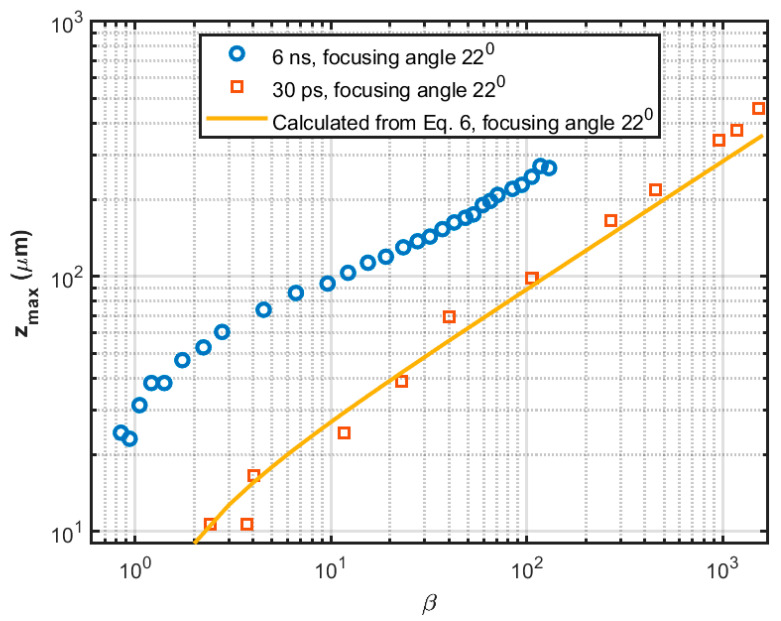
Comparison of the calculated plasma length using Equation (6) to the experimentally measured values in [55], while using 6 ns and 30 ps pulses with focusing angle 22° using 1064 nm Nd:YAG laser.

**Figure 3 sensors-22-04224-f003:**
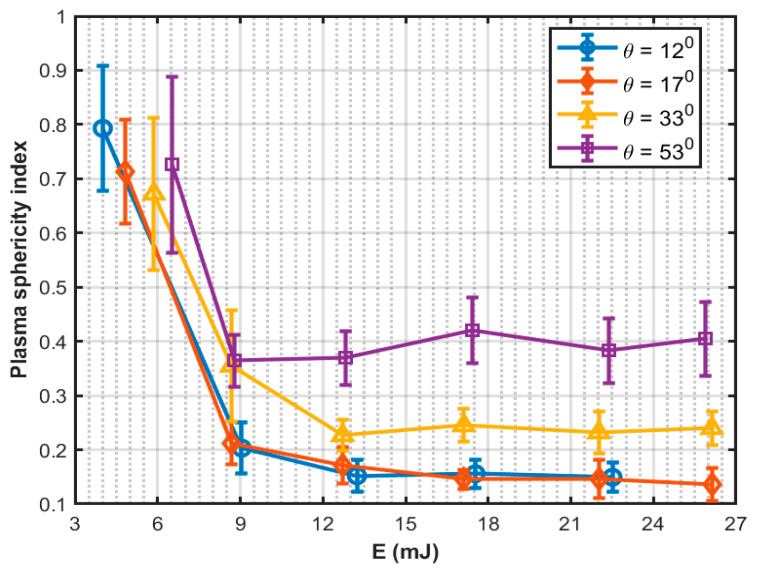
Plasma sphericity index with laser pulse energy and focusing angle, redrawn from [62].

**Figure 4 sensors-22-04224-f004:**
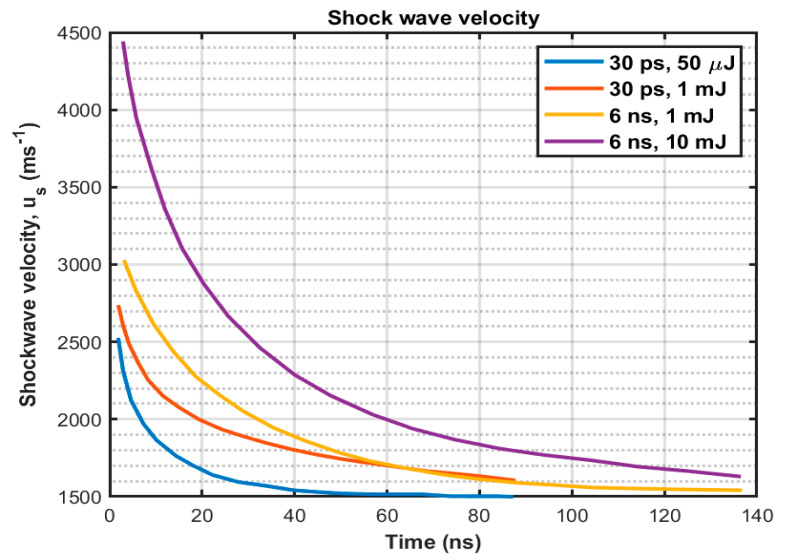
Experimentally determined shockwave velocity for 6 ns and 30 ps pulse duration, redrawn from [63].

**Figure 5 sensors-22-04224-f005:**
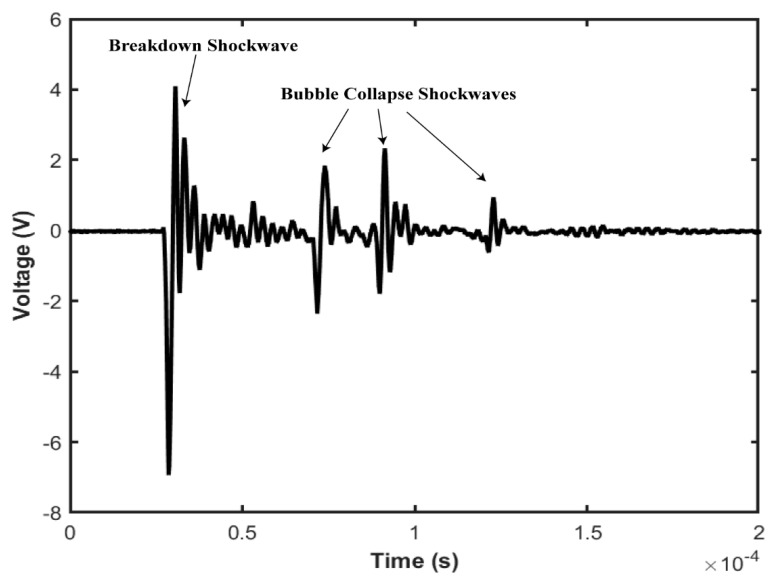
Optical breakdown generated shockwave and bubble expand–collapse shockwaves generated by a single laser pulse focused onto the water. The first transient is the acoustic shock wave generated by the optical breakdown of the water medium and the subsequent three transients are generated due to several bubble collapse and expansion cycles.

**Figure 6 sensors-22-04224-f006:**
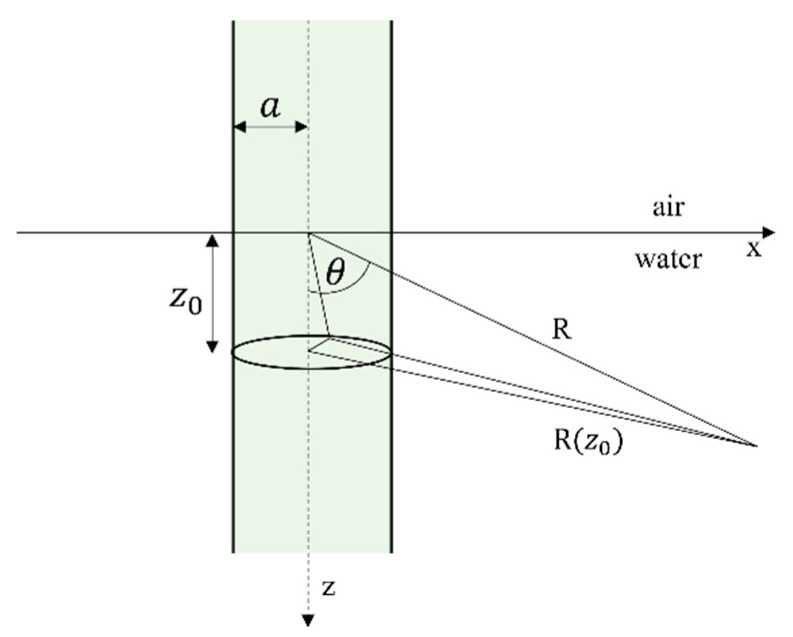
General geometry for linear optoacoustic mechanism (Redrawn from [66]).

**Figure 7 sensors-22-04224-f007:**
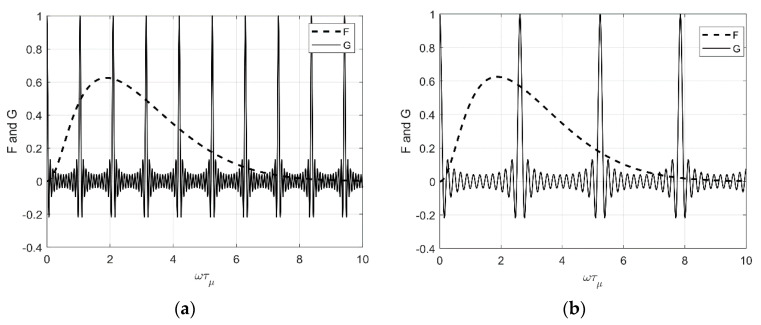
Power spectral response of optoacoustic pressure wave for (**a**) *S* = 0.5, *N* = 25, *δ* = 6 and (**b**) *S* = 0.5, *N* = 25, *δ* = 2.4.

**Figure 8 sensors-22-04224-f008:**
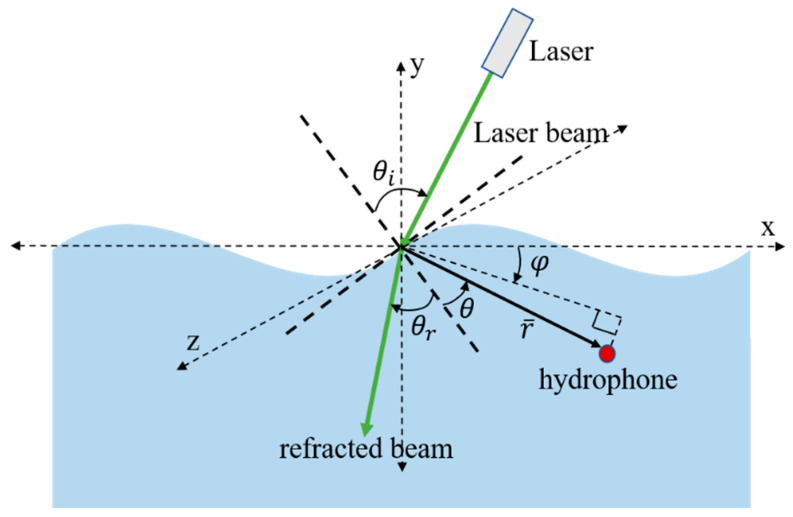
General geometry for linear optoacoustic communication from air to underwater for a wavy water surface.

**Figure 9 sensors-22-04224-f009:**
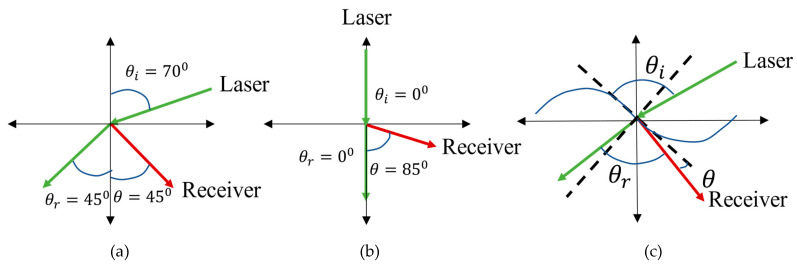
Optimum position of laser beam incident angle and receiver observation angle for flat water surface, (**a**,**b**) and wavy water surface (**c**). The figure has been redrawn from [67].

**Figure 10 sensors-22-04224-f010:**
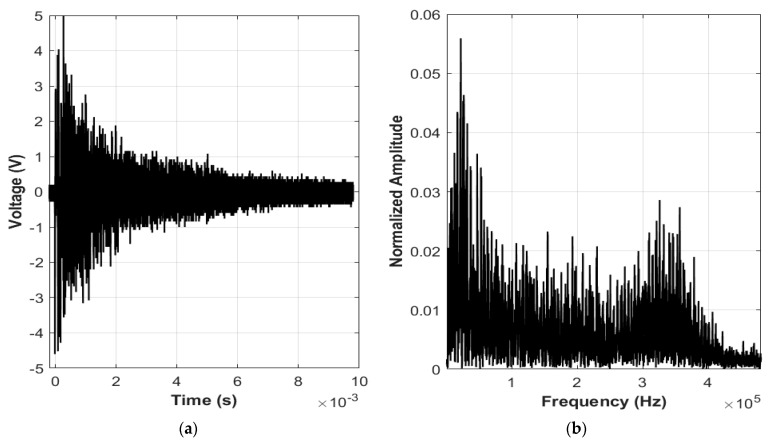
(**a**) Acoustic signal generated from a single laser pulse and (**b**) frequency spectrum of that acoustic signal.

**Figure 11 sensors-22-04224-f011:**
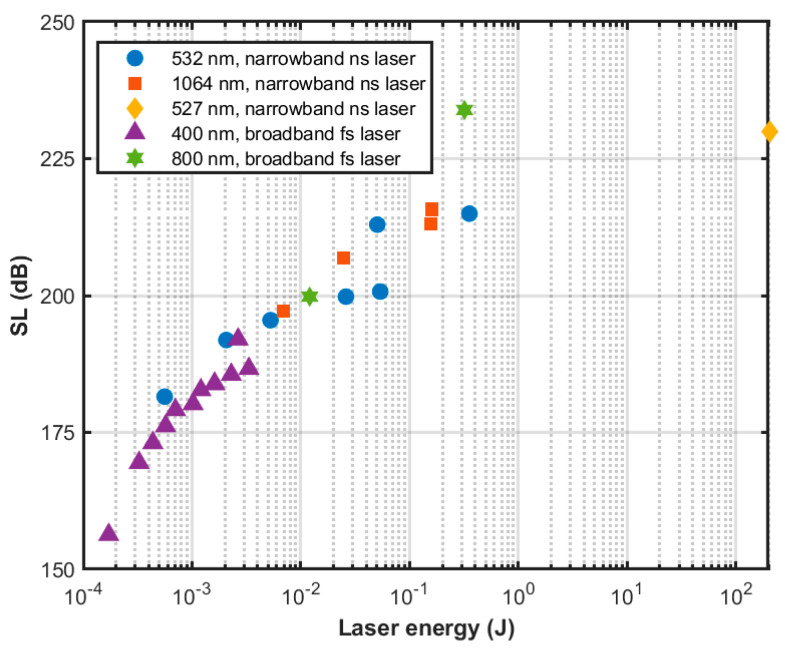
Acoustic source level as a function of laser pulse energy for non-linear photoacoustic generation with narrowband and broadband lasers.

**Figure 12 sensors-22-04224-f012:**
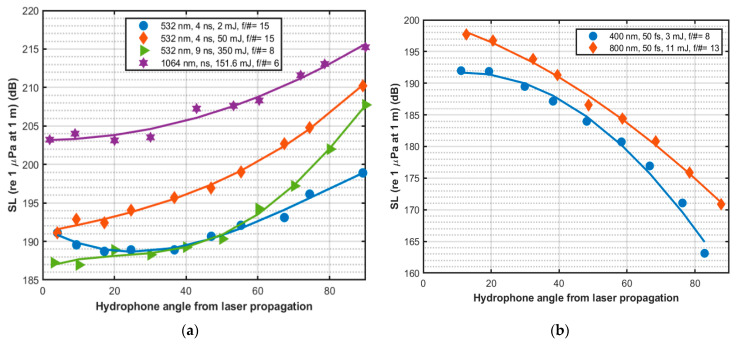
Acoustic source directionality for (**a**) narrowband nanosecond laser and (**b**) broadband femtosecond laser.

**Figure 13 sensors-22-04224-f013:**
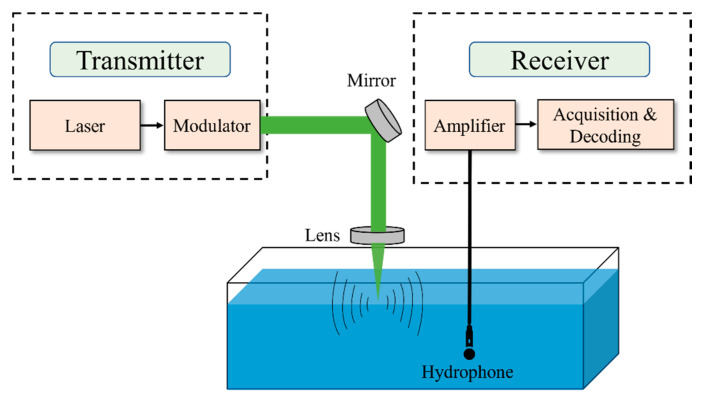
General block diagram of air to underwater photoacoustic communication.

**Figure 14 sensors-22-04224-f014:**
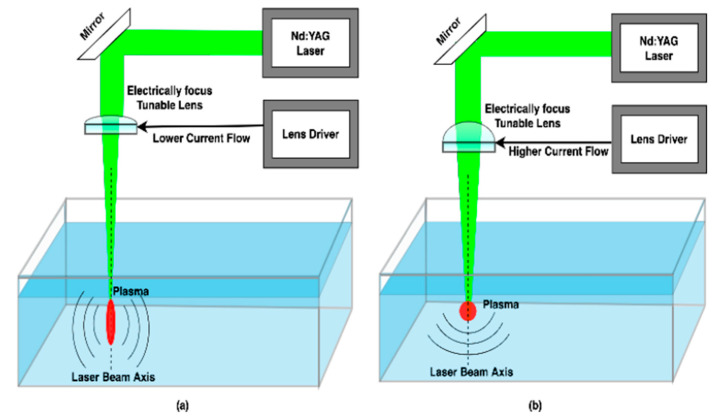
The use of electronically focus-tunable lens to create (**a**) cylindrical and (**b**) spherical shape plasma.

**Table 1 sensors-22-04224-t001:** Experimental optical breakdown thresholds for water for different pulse durations, wavelengths, and focusing angles. The spot size is experimentally measured while the *f*-number is calculated using Equation (4).

$\mathrm{Pulse}\;\mathrm{Duration},\;\tau_{L}$	Wavelength, λ(nm)	Focusing Angle, θ	*f*- Number, f/#	$\mathrm{Measured}\;\mathrm{Spot}\;\mathrm{Diameter},\;2\omega_{0}$ (μm)	Energy Threshold,$E_{th}$ (μJ)	$\mathrm{Irradiance}\;\mathrm{Threshold},\;I_{th}(10^{11}\;{\mathrm{Wcm}}^{-2})$	Source
76 ns	750	19°	2.99	20	5497.78	0.23	[54]
6 ns	1064	32°	1.74	5.39 ± 0.60	89.9 ± 0.8	0.66 ± 0.15	[55]
6 ns	1064	22°	2.57	7.66 ± 0.31	140.6 ± 1.3	0.51 ± 0.04	[55]
6 ns	1064	8°	7.15	11.53 ± 0.42	493.4 ± 11.3	0.79 ± 0.06	[55]
6 ns	1064	5.4°	10.60	14.57 ± 0.60	1082.0 ± 22.0	1.08 ± 0.09	[55]
6 ns	1064	1.8°	31.83	47.78 ± 1.14	8022.9 ±283.8	0.75 ± 0.08	[55]
6 ns	532	22°	2.57	5.31 ± 0.23	38.5 ± 0.6	0.29 ± 0.03	[55]
60 ps	532	16.7°	3.41	7.2	10.0	4.07	[56]
60 ps	532	13°	4.39	5.6	4.13	2.8	[54]
30 ps	1064	28°	2.00	4.60 ± 0.48	2.30 ± 0.01	4.61 ± 0.96	[55]
30 ps	1064	22°	2.57	4.72 ± 0.38	2.38 ± 0.01	4.53 ± 0.73	[55]
30 ps	1064	14°	4.07	5.80 ± 0.34	5.0 ± 0.07	6.31 ± 0.74	[55]
30 ps	1064	8.5°	6.73	9.60 ± 0.42	9.85 ± 0.15	4.53 ± 0.40	[55]
30 ps	1064	4°	14.32	19.54 ± 0.36	33.29 ± 0.39	3.70 ± 0.14	[55]
30 ps	1064	1.7°	33.70	47.74 ± 3.12	178.54 ± 2.62	3.32 ± 0.44	[55]
30 ps	532	22°	2.57	3.38 ± 0.32	1.01 ± 0.001	3.75 ± 0.72	[55]
3 ps	580	16.7°	3.41	10.8	1.1	4.15	[56]
3 ps	580	16°	3.56	5.0	0.51	8.5	[54]
300 fs	580	16.7°	3.41	10.8	1.0	36.0	[56]
300 fs	580	16°	3.56	5	0.27	47.6	[54]
125 fs	580	16.7°	3.41	10.8	0.35	30.6	[56]
100 fs	580	16°	3.56	4.4	0.17	111.0	[54]

**Table 2 sensors-22-04224-t002:** Experimental results of range and sound pressure level for different modulation techniques using linear photoacoustic [39].

Wave Shape	Laser Energy (J)	Experimental SL (dB re 1 μPa)	Experimental Range (M)
10-kHz Chopper	30	113	100
10-kHz Sinewave	25	111.8	78
Pulse Train	25	112.1	80
4-Bit PSK	25	106	15
11-Tone MFSK	25	101.1	23
5-Tone MFSK	25	100.7	22
2-Tone MFSK	25	104.4	33
2-Tone FSK	25	104.9	25

## Data Availability

Not applicable.

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
