# Peer review of "Cross-Medium Photoacoustic Communications: Challenges, and State of the Art"

_sensors, 2022, doi:10.3390/s22114224_

Round 1

Reviewer 1 Report

This paper investigates the Cross-Medium Photoacoustic Communications in terms of challenges and state-of-the-art. It is shown that the studied methods are effective to some degree. The paper is well organized and timely. Also, the paper presents extensive simulation results to verify the efficiency of the studied methods.

Some of my concerns are as the following.

  1. The advantages of interfacing the underwater networks to remote entities can be highlighted for a better understanding of this paper.
  2. Section I. The main contributions of this paper can be listed by points for a better understanding of this paper.
  3. Section II. It seems all the parameters in this paper are denoted by bold characters, which is not so common and may confuse readers when introducing both the scalers and the vectors.
  4. Section II. The system model is not clear. In particular, the factor of channel fading should be considered in the system model.
  5. How to obtain the channel coefficients should be discussed. Some channel estimation schemes, e.g., the works of “deep residual learning for channel estimation in intelligent reflecting surface-assisted multi-user communications”, and “Learning the MMSE channel estimator”, can be added for discussion to improve the readability of this paper.
  6. The advantages of the possible application examples can be added for discussion.

Reviewer 2 Report

The paper introduced the cross-medium photoacoustic communications and discussed technical challenges. In this manuscript, the principle of photoacoustic signal generation based on the photoacoustic (PA) effect is described. Based on the energy density and irradiance imparted to the medium, linear and nonlinear photoacoustic properties are analyzed respectively. The challenge and current state of PA-based communication and localization have also been presented. I recommend this paper to be published after some comments below are addressed.

  1. The author mentioned that “The photoacoustic (PA) effect is the formation of acoustic signals following the light absorption of the medium.” What are the requirements of the photoacoustic effect for the characteristics of the medium, the wavelength and the intensity of the light?
  2. The energy-transfer mechanism of the photoacoustic effect can be subdivided into a linear and a nonlinear domain. What’s the difference between linear and nonlinear process in principle and mathematical derivation?
  3. In Figure 2, the calculated plasma length from Eq. (6) is nearly identical to the experimentally measured values for picosecond pulses but isn’t as close for nanosecond pulses. And the author also mentioned that “the dependence of laser pulse duration is implicitly contained in Eq. (5) and (6).” What’s the relationship between plasma length and the normalized laser pulse energy under different pulse widths on earth?
  4. In Figure 5, why each bubble expansion and collapse shockwave has less peak pressure compared with the breakdown shockwave?

Reviewer 3 Report

This survey paper provided an overview of the state of the art in air-to-underwater communication using the photoacoustic mechanism.  This is a very important research topic and the paper offered details in introducing theories behind the mechanism. The only negative part could be the very brief coverage of the "PA-enabled underwater localization" (section 5.3) and the over simplified conclusion (section 6).   

Reviewer 4 Report

The scientific basics are only presented in a very rudimentary way, especially in connection with the incidence at sea, so I ask for a revision, otherwise this state of the art report is successful and has its justification.

Reviewer 5 Report

The abstract needs to be rewritten and improved. 

Figure 1 does not convey the idea well and is not explained. 

Some recent underwater acoustic communication papers must be cited.

There are many grammatic mistakes here and there, the document needs a thorough revision. 

Table 3 should have more parameters and probably a table of all the techniques compared in different sections.
